# Non-Syndromic and Syndromic Defects in Children with Extracranial Germ Cell Tumors: Data of 2610 Children Registered with the German MAKEI 96/MAHO 98 Registry Compared to the General Population

**DOI:** 10.3390/cancers16112157

**Published:** 2024-06-06

**Authors:** Judit H. Schultewolter, Anke Rissmann, Dietrich von Schweinitz, Michael Frühwald, Claudia Blattmann, Lars Fischer, Björn Sönke Lange, Rüdiger Wessalowski, Birgit Fröhlich, Wolfgang Behnisch, Irene Schmid, Harald Reinhard, Matthias Dürken, Patrick Hundsdörfer, Martin Heimbrodt, Christian Vokuhl, Stefan Schönberger, Dominik T. Schneider, Guido Seitz, Leendert Looijenga, Ulrich Göbel, Rüdiger von Kries, Heiko Reutter, Gabriele Calaminus

**Affiliations:** 1University Bonn, Venusberg Campus 1, 53127 Bonn, Germany; 2Malformation Monitoring Centre Saxony-Anhalt, Medical Faculty, Otto-von-Guericke University, 39106 Magdeburg, Germany; anke.rissmann@med.ovgu.de; 3Dr. von Haunersches Kinderspital, Department of Paediatric Surgery, University of Munich, 80539 Munich, Germany; dietrich_schweinitz@hotmail.de; 4Department of Pediatric and Adolescent Medicine, University Medical Center Augsburg, 86159 Augsburg, Germany; michael.fruehwald@klinikum-augsburg.de; 5Centre for Childhood, Adolescents and Female Medicine, Paediatrics 5 (Oncology, Hämatology, Immunology), Olgahospital Klinikum Stuttgart, 70174 Stuttgart, Germany; c.blattmann@klinikum-stuttgart.de; 6Clinic for Childhood and Adolescent Medicine, Paediatric Oncology, University Hospital Leipzig (Universitätsklinikum Leipzig AöR), 04103 Leipzig, Germany; lars.fischer@medizin.uni-leipzig.de; 7Clinic for Childhood and Adolescent Medicine, Paediatric Haematology and Oncology, University Hospital Dresden, 01307 Dresden, Germany; bjoernsoenke.lange@ukdd.de; 8Clinic for Paediatric Hematology, Oncology and Immunology, University Childrens Hospital Düsseldorf, 40225 Düsseldorf, Germany; wessalowski@med.uni-duesseldorf.de (R.W.); goebelu@arcor.de (U.G.); 9Clinic for Paediatric Hematology and Oncology, University of Münster, 48149 Münster, Germany; birgit.froehlich@ukmuenster.de; 10Department of Paediatric Haematology and Oncology, University Childrens Hospital Heidelberg, 69120 Heidelberg, Germany; wolfgang.behnisch@med.uni-heidelberg.de; 11Dr. von Haunersches Kinderspital, Department of Paediatric Haematology and Oncology, University of Munich, 80539 Munich, Germany; irene.schmid@med.uni-muenchen.de; 12Department of Paediatric Haematology and Oncology, Asklepios Hospital Sankt Augustin, 53757 St. Augustin, Germany; h.reinhard@asklepios.com; 13Clinic for Childhood and Adolescent Medicine, Paediatric Haematology and Oncology, Medical Faculty Mannheim, University of Heidelberg, 69117 Heidelberg, Germany; matthias.duerken@umm.de; 14Clinic for Childhood and Adolescent Medicine, Oncology Haematology, HELIOS Clinic Berlin-Buch, 13125 Berlin, Germany; patrick.hundsdoerfer@helios-gesundheit.de; 15Department of Pediatric Hematology and Oncology, University Hospital Bonn, 53127 Bonn, Germany; martin.heimbrodt@ukbonn.de (M.H.); gabriele.calaminus@ukbonn.de (G.C.); 16Department of Pathology, Section Paidopathology, University Hospital Bonn, 53127 Bonn, Germany; christian.vokuhl@ukbonn.de; 17Department of Pediatric Hematology and Oncology, University Hospital Essen, University of Duisburg-Essen, 45147 Essen, Germany; stefan.schoenberger@uk-essen.de; 18Clinic of Paediatrics, Klinikum Dortmund, University Witten/Herdecke, 58448 Witten, Germany; dominik.schneider@klinikumdo.de; 19Department of Pediatric Surgery and Urology, University Hospital Giessen-Marburg, Campus Marburg, 35037 Marburg, Germany; guido.seitz@uk-gm.de; 20Department of Pediatric Surgery, University Hospital Giessen-Marburg, Campus Giessen, 35392 Giessen, Germany; 21Princess Máxima Center for Pediatric Oncology, 3584 Utrecht, The Netherlands; l.h.j.looijenga-2@prinsesmaximacentrum.nl; 22Division of Epidemiology, Institute of Social Pediatrics and Adolescent Medicine, LMU Munich, 80539 Munich, Germany; rvkries@mail.de; 23Department of Pediatrics and Adolescent Medicine, University Hospital Erlangen, 91054 Erlangen, Germany; heiko.reutter@uk-erlangen.de

**Keywords:** germ cell tumor, syndrome, non-syndromic defect, developmental tumor, Swyer syndrome, Currarino syndrome, MAKEI 96/MAHO 98

## Abstract

**Simple Summary:**

Our findings confirm the role of developmental aspects in extracranial germ cell tumors (eGCTs). Strong associations between syndromes or congenital anomalies and eGCTs may hint at the need for research to identify potential causes and mechanisms. Strong effects were observed for Swyer and Currarino syndrome. The strengths of the associations emphasize the need for further elaboration on presently discussed pathophysiological mechanisms. For other suggested risk factors such as Klinefelter, Turner, and Down syndrome, the strength of risk appears to be substantially low and may be missed because of diagnostic bias for the first two or lack of power for the latter. A number of further syndromes were identified in single cases. Most of these pertain to girls.

**Abstract:**

GCTs are developmental tumors and are likely to reflect ontogenetic and teratogenetic determinants. The objective of this study was to identify syndromes with or without congenital anomalies and non-syndromic defects as potential risk factors. Patients with extracranial GCTs (eGCTs) registered in MAKEI 96/MAHO 98 between 1996 and 2017 were included. According to Teilum’s holistic concept, malignant and benign teratomas were registered. We used a case–control study design with Orphanet as a reference group for syndromic defects and the Mainz birth registry (EUROCAT) for congenital anomalies at birth. Co-occurring genetic syndromes and/or congenital anomalies were assessed accordingly. Odds ratios and 95% confidence intervals were calculated and *p*-values for Fisher’s exact test with Bonferroni correction if needed. A strong association was confirmed for Swyer (OR 338.6, 95% CI 43.7–2623.6) and Currarino syndrome (OR 34.2, 95% CI 13.2–88.6). We additionally found 16 isolated cases of eGCT with a wide range of syndromes. However, these were not found to be significantly associated following Bonferroni correction. Most of these cases pertained to girls. Regarding non-syndromic defects, no association with eGCTs could be identified. In our study, we confirmed a strong association for Swyer and Currarino syndromes with additional congenital anomalies.

## 1. Introduction

Extracranial pediatric germ cell tumors (eGCTs), a model of developmental tumorgenesis, are rare neoplasms occurring in different sites and ages [1]. According to data from the DKKR (Deutsches Kinderkrebsregister) from 1987 to 2011, the incidence rate is 4.8 per million (children aged <15 years). For girls, the incidence is 5.3, and for boys, it is 4.4 per million [2]. In a study from 2023, the 5-year survival rate (EFS) and overall survival were 82.33% (95% CI, 77.32% and 86.62%) and 94.13% (95% CI, 90.02% and 96.69%), respectively [3]. These tumors arise from primordial germ cells and derivatives and are localized in the gonads or in extragonadal tissues along the body midline, both extra- and intracranially [4,5]. The preferred regions of origin of eGCTs are the gonads (testis and ovary), sacrococcygeal region, and mediastinum [6]. eGCTs are classified according to originally Teilum with respect to their histological degree of embryonic differentiation (teratoma or embryonal carcinoma), extraembryonic differentiation (yolk sac tumor or choriocarcinoma), and germinal differentiation (seminoma in male patients, dysgerminoma in female patients, and germinoma when occurring in the CNS) [4]. Mixed forms with more than one histological subtype are possible, especially beyond childhood (ICD-11—Mortality and Morbidity Statistics (who.int (accessed on 18 April 2024)).

Some risk factors have been suggested, including genetic syndromes such as Swyer syndrome [7,8], Down syndrome [9,10,11], Currarino syndrome defined by the triad of a presacral mass, sacral and anorectal congenital anomalies [12], or Klinefelter syndrome [13,14]. Furthermore, the occurrence of eGCTs has been associated with non-syndromic defects as, e.g., cryptorchidism, inguinal hernias, congenital heart defects, and skeletal congenital anomalies [15,16]. A recent publication suggested that birth defects might only be related to eGCTs if related to syndromes (syndromic defects) but not for non-syndromic defects.

We analyzed cases of the German MAKEI 96/MAHO 98 registry providing data on eGCT. To identify potentially related risks in a broader range of syndromes, we identified all cases with any reported syndrome in MAKEI/MAHO and compared their prevalence to that reported in Orphanet. Additionally, we identified cases with reported non-syndromic defects and addressed their overall potential risk for any eGCT and by specific tumor site using a German birth cohort. We performed an aggregated data analysis. The objective was to confirm or refuse already known risk factors regarding syndromes with or without associated congenital anomalies, to identify potentially new syndromes associated with eGCTs, and to assess a potential role for non-syndromic defects.

## 2. Materials and Methods

The MAKEI 96 registry was conducted in accordance with the Declaration of Helsinki and was approved by the Ethics Committee of Heinrich-Heine-University, Düsseldorf (study number 837, decision of 22 August 1995). All participants provided their written informed consent. Informed consent was given by the respective legal guardians of the affected children. All patients had an eGCT. In the patient data evaluated here, cases with an additional non-oncologic diagnosis (genetic syndromes or congenital anomalies) were assessed in detail.

### 2.1. Study Population

The MAKEI 96/MAHO 98 registry comprises a large data collection of children and adolescents with an eGCT (*n* = 2610) [17,18]. Patients were prospectively enrolled in the registry over a 20-year period. Data were collected in collaboration with centers of the Society for Pediatric Oncology and Hematology (GPOH). Between January 1996 and October 2017, a total of 2610 patients with a newly diagnosed eGCT were prospectively registered by 62 pediatric oncology units in Germany, Austria, and Switzerland. Presacral, sacral, and coccygeal tumors were grouped together in the sacrococcygeal group. The presacral mass occurred as part of the Currarino syndrome.

In the current study, all patients with documented associated genetic syndromes and congenital anomalies of the genitourinary tract, the heart, the digestive system, the musculoskeletal system, and the central nervous system were identified. During the initial survey following the registration of patients, epidemiologic data were collected by the MAKEI 96/MAHO 98 registry. In case of incomplete reporting, the hospitals were reminded until presumed complete reporting was achieved. Basic demographic data were also documented. The CRF included information on anatomical site of origin, symptoms, diagnosis, tumor markers, and histology. Furthermore, non-oncologic diagnoses such as underlying genetic syndromes, chronic diseases, and congenital anomalies were documented.

### 2.2. Case Definitions

Syndromes categorized according to Orphanet in MAKEI/MAHO were identified. Verification of the diagnoses by molecular genetic analysis was beyond the scope of the MAKEI/MAHO registry. For analysis, we used any syndrome irrespective of any potential additional congenital anomalies.

The diagnosis of congenital anomalies was based on clinical aspects, ultrasound, or CT/MRI imaging (in cases). In accordance with EUROCAT, non-syndromic defects were grouped according to their localization [19,20,21]. We additionally considered the associated type of tumor and potential syndrome.

### 2.3. Definition of Exposure

In accordance with Schraw et al. [22], we distinguished the effects of syndromes and isolated non-syndromic defects. In cases with an associated syndrome, we could not distinguish those potential additional non-syndromic defects because of the aggregated data structure regarding controls.

### 2.4. Reference Population

To assess the association with syndromes, we used the respective prevalence of genetic syndromes in the general population as documented in Orphanet. The prevalence of genetic syndromes in the general population was extracted from the Orphanet database (https://www.orpha.net/, accessed on 18 April 2024).

The principle of the case–control study is to compare the prevalence of cases (MAKEI 96/MAHO 98) with or without genetic syndromes to controls from the general population. The prevalence of non-syndromic defects in the general population was estimated from the Mainz malformation registry with a complete, extensive, and standardized assessment of birth defects in the first week of life, including ultrasound of the urogenital system at birth. The data were extracted from the EUROCAT registry for children born between 1996 and 2017 (http://www.eurocat-network.eu/accessprevalencedata/prevalencetables, accessed on 18 April 2024). In the Mainz registry, cases are highly likely to reflect the general population [23].

### 2.5. Statistical Analysis

Because the registries provided only aggregated data, case data were aggregated accordingly using Excel files (Version 2405). These aggregated data were entered into stat calc to calculate the odds ratios and 95% confidence intervals. Stat calc is a public domain app sponsored and provided by CDC (https://www.cdc.gov/epiinfo/user-guide/statcalc/statcalcintro.html, accessed on 18 April 2024). Stat calc also provides formal statistical testing using different methods. We used the *p*-values for Fisher’s exact test with additional Bonferroni correction for multiple testing regarding potential new syndromes associated with eGCTs. To assess whether the number of cases was sufficient to detect the observed effect size in previously reported associations with a power of at least 80%, we used the calculation of power with the power and sample size program (https://vbiostatps.app.vumc.org/ps/ accessed on 18 April 2024) in a post hoc analysis.

## 3. Results

### 3.1. Overall Frequency of Registered eGCTs

The most frequent eGCT in the total MAKEI 96/MAHO 98 registry sample of 2610 patients at the time of analysis were those of the ovaries (47%, *n* = 1232), followed by those of the sacrococcygeal region (21%, *n* = 540), the testis (17%, *n* = 447), and the mediastinum (4%, *n* = 114). A total of 277 tumors were found in other anatomical structures of the body midline (11%), neck, eye, or retroperitoneum. Of the 2610 patients with an eGCT, 754 were male (28.9%), and 1856 were female (71.1%). In comparison, in the group of patients with an eGCT and an additional non-oncologic diagnosis, 44 were male (45.9%), and 51 were female (54.1%).

### 3.2. Location and Histology of eGCTs in Patients with Genetic Syndromes

Gonadal GCTs: 34 patients with genetic syndromes were found in combination with a gonadal GCT (26 of 1232 ovarian GCTs and 7 of 447 testicular GCTs). A percentage of 62% of the 26 ovarian GCTs were dysgerminomas, followed by their precursor lesion gonadoblastomas (15%). When analyzing patients with a genetic syndrome and a testicular GCT, four (57%) presented with mixed eGCTs. In addition, one case each of seminoma, embryonal carcinoma, and embryonal carcinoma with a yolk sac tumor was detected.

Extragonadal GCTs: Nine patients with a tumor in the sacrococcygeal region (9 of 540 tumors of the sacrococcygeal region) were documented. In the group of sacrococcygeal tumors, only teratomas were found, and one of them was combined with a yolk sac tumor. Three patients with a genetic syndrome had a mediastinal GCT (3 of 114) comprising one choriocarcinoma and two mixed eGCTs (one composed of a yolk sac tumor and a teratoma, the other composed of a choriocarcinoma, a yolk sac tumor, and a teratoma). Among patients with an eGCT of the retroperitoneum, 3 patients (out of 277) had a genetic syndromic background comprising Down syndrome, neurofibromatosis type 1 and one patient with intellectual disability, thoraco-lumbar scoliosis, and tethered cord. Among these, two patients had teratomas, and one had a mixed eGCT (embryo carcinoma, choriocarcinoma, and yolk sac tumor).

### 3.3. eGCT and Associated Genetic Syndromes

Of the 95 patients with an additional non-oncologic diagnosis, 48 had a syndrome. Most syndromes recorded in the MAKEI/MAHO registry could be linked to a specific tumor (Table 1), except for Down syndrome, which was found in connection with eGCTs of different localizations. Most underlying genetic syndromes were found in the group of patients with an ovarian eGCT (*n* = 26). In terms of age, most eGCTs in patients with an additional syndrome were diagnosed in the prepubertal or pubertal period. Only in the group of sacrococcygeal GCTs was the peak incidence in the first year of life. The majority of eGCTs within this group were malignant; only in patients with Currarino syndrome were benign eGCT tumors predominated. 

Overall associated syndromes were only rarely detected in the study group with eGCTs (1.8%). Regarding any syndrome registered in MAKEI 96/MAHO 98, the odds ratio was 2.9 (95% CI 2.2–3.8) (Figure 1). The previously described correlations between Swyer and Currarino syndromes and the occurrence of eGCTs were confirmed. The strength of the effect was substantial, with an OR for Swyer syndrome above 100 and 34 for Currarino syndrome, respectively, with 95% CI clearly excluding 1. For Down syndrome, the OR was 1.5 with a 95% CI including 1 (95% CI 0.2–12.4) despite a considerable number of cases. The power calculation in a post hoc analysis showed that the power to detect the observed effect with an alpha of <0.05 was only 0.08. For Klinefelter and Turner syndromes [41], we failed to identify a significant association [42]. The effect estimates for the known syndromes did not require Bonferroni correction because of a priori hypothesis (Figure 1).

Interestingly, 16 new syndromes were found in children with eGCTs, and most of them were female (75%, *n* = 12). However, these were only isolated cases, never more than one. Compared to the registry, these yielded low *p*-values for Fisher’s exact and for some high OR, which were driven by the extreme paucity of such syndromes in the general population. Following Bonferroni correction for multiple testing, the respective *p*-values were clearly above the Bonferroni adjusted *p*-value of 0.00238, however (see Appendix A). A remarkable finding, however, is the fact that most cases were tied to girls. While this is evident for the Triple X syndrome, the sex distribution of the other syndromes was either equal or slightly shifted to male or female patients. On average, the prevalence for girls in the general population was 42% for the syndromes mentioned (https://www.orpha.net/, accessed on 18 April 2024).

### 3.4. eGCT and Associated Malformations

In the MAKEI 96/MAHO 98 registry, 47 patients with a non-syndromic defect were recorded. Non-syndromic defects occurred in connection with eGCTs of all localizations. Most were described in the group of sacrococcygeal GCTs (*n* = 18) (Table 2). Urogenital malformations accounted for the largest proportion. The age distribution at the time of tumor diagnosis also showed a peak postnatally for the sacrococcygeal GCTs and prepubertal or pubertal for the other tumor localizations.

For all non-syndromic defects in the MAKEI/MAHO registry, compared to the data from the Mainz registry, the OR was 0.6 (95% CI 0.5–0.9) (Figure 2). Thus, an overall risk related to isolated congenital anomalies appears highly unlikely. To assess whether the local vicinity of tumors and congenital anomalies might be related to a higher risk, we performed a subgroup analysis. For most subgroups, there was no evidence, and following Bonferroni correction (Bonferroni corrected *p*-value 0.005), none of the associations reached significance. Overall, no strong effect sizes were found.

## 4. Discussion

The main finding of our paper was the confirmation of the role of several syndromes in accounting for the risk of an eGCT. We observed high ORs for both Swyer and Currarino syndromes. Regarding non-syndromic defects, no overall association was found. Subgroup analysis might suggest a hint at a potential role for non-syndromic defects close to the tumor site. 

The identification of risk factors requires a control population and cautious interpretation of statistical findings. In the literature, few papers used a control group describing potential associations of syndromes and the later emergence of eGCTs [43,44,45,46,47]. Whereas some previous papers only reported associations between tumor and syndrome in cases [48,49,50,51], a recent paper with a control group allowing for individual case–control analysis with adjustment confirmed a strong statistical association for Klinefelter with comparison to controls, but not for Turner and Down syndrome [22]. Our data confirmed the latter. Regarding our data, the estimate for Klinefelter syndrome was likely, however, to reflect diagnostic bias. Unlike in other registers [22] no systematic screening for Klinefelter was performed in our MAKEI/MAHO patients. Similarly, no systematic screening for Turner syndrome was performed in eGCT cases in MAKEI/MAHO. As many of the eGCTs occur prepubertal or in early puberty, such cases may well be missed because of the later occurrence of the lead symptoms. Down syndrome, in contrast, is most likely to be diagnosed early in life. Additionally, some biological probability for causality was described the in the literature. The result of a study by Cools et al. (2006) was a longer expression of OCT3/4, PLAP, and c-KIT, biomarkers for delayed primordial germ cell maturation, linking it to the development of the precursor of a malignant eGCT, via either GCNIS or gonadoblastioma, in male fetuses with trisomy 21 [52]. The estimate for the strength of the effect, however, was low (OR 1.5) with a wide 95% CI clearly including one. In a post hoc power analysis, we found that with the observed case numbers the power to detect such a small effect size was only very low (0.08).

Regarding the presumed new syndromes, our data point to the need for a very careful interpretation of statistics in case of isolated findings of syndromes in eGCT cases. Although effect estimates based on comparison with control groups may appear huge, the risk is far from proven. In the case of many “new associations”, it is important to perform at least a Bonferroni correction. On the other hand, a causative association is difficult to rule out, especially when only isolated cases are identified. Although a role for the individual syndromes could not be established, confinement to girls is striking. Among patients with gonadal eGCT, associations with genetic syndromes based on disorders of sexual development have been described previously. About 1 in 4500 births (0.02%) is affected by a disorder of sexual development [20]. Several publications describe an increased risk of developing an eGCT in individuals with gonadal dysgenesis or in patients with Turner syndrome and Y chromosomal material [7,53,54,55,56,57,58]. The MAKEI 96/MAHO 98 registry comprised 18 patients with gonadal dysgenesis who had an underlying genetic syndrome (0.69%). Among patients with published results with disorders of sexual development, gonadoblastoma is diagnosed in 4.7–25% of cases. Of these, 50% develop a dysgerminoma [15,59,60,61]. These histology types were prominent in the 18 patients of the MAKEI 96/MAHO 98 registry (15 of 18 patients with pure gonadoblastoma and 11 of 18 patients with gonadoblastoma and dysgerminoma). eGCTs most commonly occur in gonadal dysgenesis in association with Swyer syndrome [62]. Various molecular biomarkers, such as miR-371a-3p, were discovered in patients with eGCT and disorders of sexual development [63].

Mutations of the MNX1 gene have been found in patients with Currarino syndrome, and further molecular genetic correlations are being sought [64].

For non-syndromic defects, we failed to identify an increased risk for eGCTs in accordance with Schraw et al. [22]. A subgroup analysis linking tumor site and congenital anomaly did not yield significant results either.

Confirmation of high risks for Currarino and Swyer syndrome might suggest a cause for screening in these syndromes because in children with these syndromes, the a priori probability of GCTs might be high enough to allow for screening with acceptable positive predictive values. Unfortunately, however, most cases of Currarino syndrome occur in the first year of life when the test is not specific. For Swyer syndrome with pure dysgerminomas, a screening is not useful because of the absence of known tumor markers such as AFP and beta-HCG.

### Strengths and Limitations

The MAKEI 96/MAHO 98 registry is very complete regarding the recording of eGCT.

Beyond the presumed complete eGCT registration for Germany in MAKEI 96/MAHO 98 and comparison to a standard control group (Orphanet), the strength of our paper pertains to statistical analysis. We provide the mandatory Bonferroni correction if needed and show that an apparently strong effect suggested for one case only in extremely rare conditions may well be spurious. Such associations should only be discussed in case of possible causative pathways. Regarding congenital anomalies, the Mainz registry, like EUROCAT, aims to provide essential epidemiological information on the birth prevalence of congenital malformations and anomalies in Europe [18,19]. The strength of the Mainz registry, however, is a complete case ascertainment and a structured clinical assessment and ultrasound screening for the entire cohort. Therefore, the Mainz registry provides a unique, valid, comparator to presumed complete congenital anomaly assessment in the cases. The latter is likely because all the children with eGCT undergo extensive screening for metastasis. As with all registries, there is an issue about diagnostic and an ascertainment bias. Regarding the cases, this would account for underestimation of the effect sizes. As outlined, specific syndromes such as Turner and Klinefelter syndrome may be very plausible reasons for diagnostic bias. For other syndromes and non-syndromic defects, this may be less evident but cannot be excluded.

Previous articles on the association of an eGCT and an additional non-oncologic diagnosis were based on small data sets or case reports. As the proportion of diagnosed eGCTs increased over the observed period, we assume that this could be due to the improved diagnostic capabilities that developed over time. Still, our study has several limitations. Compared to today, fewer predispositions to tumors were known at the beginning of data collection. Not all co-occurring phenotypic features among other patients registered in the MAKEI 96/MAHO 98 registry have led to the diagnosis of a genetic syndrome. Any errors in diagnosis or coding may adversely affect the accuracy of the presented results. Furthermore, under-reporting may be an issue as patients with sacrococcygeal eGCTs; in particular, those who undergo postnatal surgery in smaller hospitals might not been registered in the MAKEI 96/MAHO 98 registry. In addition, potential candidates for the registry may have died from pathologies due to genetic syndrome or congenital anomalies before the eGCT could be diagnosed and registered. An evaluation of the success of the therapy in the reported cases was not part of this paper. In the eGCTs of the sacrococcygeal region, tumors with different relationships to the os sacrum and os coccyx were combined without considering the migration of the primordial germ cells. Further, only one leading syndromic abnormality has been documented per case. Regarding congenital anomalies, one case may be recorded for different anomalies, also affecting the statistical results. Patients who were diagnosed with eGCT and congenital anomalies shortly after birth were also included. No data was collected on premature births as a possible cause of the anomaly. Possible overlaps between the data sets of the MAKEI 96/MAHO 98 registry, EUROCAT, and the Mainz registry cannot be ruled out with certainty.

Unfortunately, we could not access the entire Mainz data to allow for individual case statistical analysis because the registration in the Mainz registry has been closed, and the involved persons have retired. This is why we could only use the Mainz registry as reported in EUROCAT.

## 5. Conclusions

In our study, we confirmed an association between Swyer and Currarino syndrome and eGCTs with high effect sizes. There is biological plausibility for these associations. Our failure to confirm an association between Klinefelter and Turner syndrome and eGCTs might be due to probable diagnostic bias of these conditions in cases. The presumed size of the association of Down syndrome with eGCT appeared to be small and thus could not be identified with the available case numbers. Case reports of associations of rare syndromes in children with eGCT are most likely to be caused by chance. Considering non-syndromic defects, no association was shown in relation to eGCTs.

## Figures and Tables

**Figure 1 cancers-16-02157-f001:**
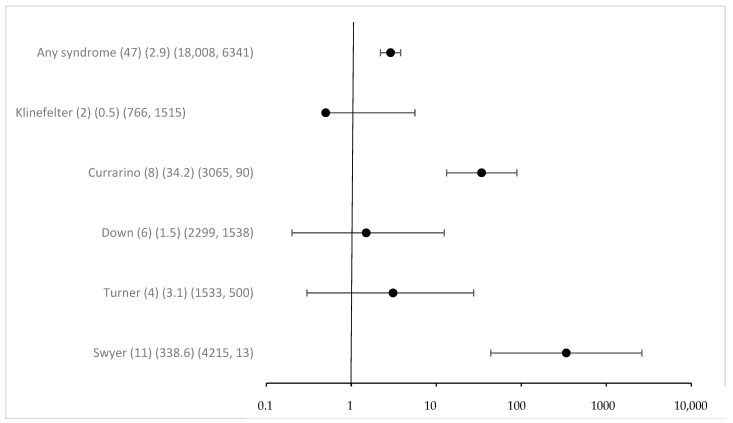
Odds ratio and 95% confidence intervals for eGCT in patients with different syndromes. Provided data (*n*) (OR) (exposure per million in cases, controls).

**Figure 2 cancers-16-02157-f002:**
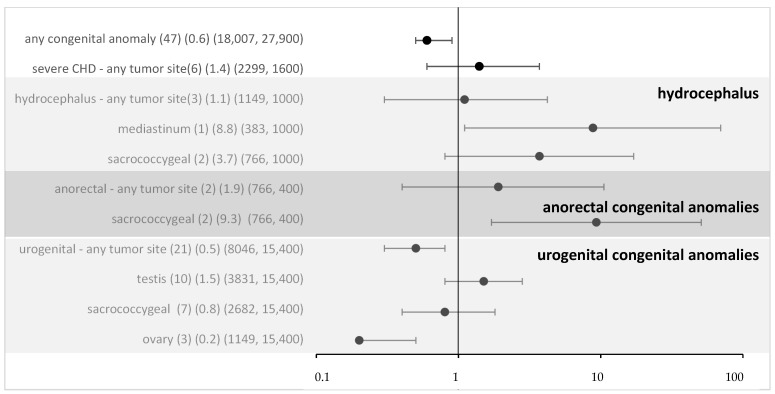
Odds ratio and 95% confidence intervals for eGCT in patients with different congenital anomalies and by tumor sites (sensitivity analysis with data from Mainz). Provided data (*n*) (OR) (exposure per million in cases, controls).

**Table 1 cancers-16-02157-t001:** Germ cell tumors are divided by localization and co-occurring syndromes (number of cases, sex, age at diagnosis, and dignity).

Tumor Localization	Syndromes	N	Sex	Median Age at Diagnosis (Range)	Malignant
Ovary	Swyer	11	f	13 years (9–16 years)	9/11
Ovary	Turner	4	f	13 years (8–19 years)	2/4
Ovary	Wolf–Hirschhorn [24]	1	f	10 years	1/1
Ovary	Campomelic dysplasia [25,26,27]	1	f	16 years	1/1
Ovary	Cowden [28]	1	f	8 years	1/1
Ovary	Denys Drash [29,30]	1	f	5 years	1/1
Ovary	Frasier [30,31]	1	f	6 years	1/1
Ovary	Imerslund–Gräsbeck [32]	1	f	14 years	1/1
Ovary	Lennox [33]	1	f	15 years	1/1
Ovary	Noonan [34]	1	f	16 years	1/1
Ovary	Poland [35]	1	f	10 years	0/1
Ovary	Proteus	1	f	15 years	0/1
Ovary	Rett [36]	1	f	13 years	1/1
Testis, mediastinum, other	Down	6	5 m/1 f	14 years (7 months–19 years)	5/6
Testis	Sturge–Weber [37]	1	m	14 years	1/1
Testis	West [38]	1	m	16 years	1/1
Testis	Cornelia de Lange [39]	1	m	15 years	1/1
Testis	Familial myoclonic dystony	1	m	14 years	1/1
Mediastinum	Klinefelter	2	m	12 years	2/2
Sacrococcygeal	Triple X	1	f	one day	1/1
Sacrococcygeal	Currarino	8	4 m/4 f	3 years (2 months–15 years)	2/8
Other	Neurofibromatosis I [40]	1	f	16 years	0/1

**Table 2 cancers-16-02157-t002:** Germ cell tumors divided by co-occurring congenital anomalies and tumor location (number of cases, sex, age at diagnosis, and dignity (number of malignant tumors of all tumors belonging to a localization)).

Tumor Localization (*n*)	Congenital Anomaly	*n*	Sex	Median Age at Diagnosis (Range)	Malignant Tumor
Ovary (10)	Urogenital	3	f	12 years (9–16 years)	2/10
	Anorectal	0			
	Hydrocephalus	0			
	Severe CHD	1	f	12 years	2/10
	Other	6	f	11 years (4–14 years)	2/10
Sacrococcygeal (18)	Urogenital	7	4 f/3 m	1 month (1 day–6 months)	3/18
	Anorectal	1	f	4 months	1/18
	Hydrocephalus	2	m	1 day	1/18
	Severe CHD	0			
	Other	8	3 f/5 m	6 months (1 day–2 years)	5/18
Testis (13)	Urogenital	10	m	7 years (3 months–16 years)	7/13
	Anorectal	0			
	Hydrocephalus	0			
	Severe CHD	2	m	9 years (1–17 years)	2/13
	Other	1	m	13 years	1/13
Mediastinum (2)	Urogenital	0			
	Anorectal	0			
	Hydrocephalus	1	m	16 years	1/2
	Severe CHD	1	m	13 years	1/2
	Other	0			
Other (4)	Urogenital	1	m	Unknown	0/4
	Anorectal	0			
	Hydrocephalus	0			
	Severe CHD	1	m	19 years	1/4
	Other	2	1 f/1 m	Unknown	2/4

## Data Availability

Restrictions apply to the availability of these data. Data were obtained from EUROCAT (Mainz registry) and are available at https://eu-rd-platform.jrc.ec.europa.eu/eurocat/eurocat-data/prevalence_en (accessed on 18 April 2024) with the permission of EUROCAT (Mainz registry).

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
