# Peer review of "Non-Syndromic and Syndromic Defects in Children with Extracranial Germ Cell Tumors: Data of 2610 Children Registered with the German MAKEI 96/MAHO 98 Registry Compared to the General Population"

_cancers, 2024, doi:10.3390/cancers16112157_

Round 1
Reviewer 1 Report
Comments and Suggestions for Authors
In this paper, the authors compared eGCT with and without some disease syndromes between German MAKEI 96/MHO 98 and general population. The following are a fews concerns and questions I have.
1. A major concern is about the study population, German MAKEI 96/MHO 98. "According to data from the DKKR (Deutsches Kinderkrebsregister) from 1987 to 2011, the incidence rate is 4.8 per million (children aged < 15 years). For girls, the incidence is 5.3, and for boys, 4.4 per million (2)". The difference between boys and girls are not very large. However, in the study population, "Of the 2,610 patients with an eGCT, 754 were male (28.9 %) and 1856 were female (71.1 %). " Almost 3 times difference between girls and boys. Why is there such a big difference between DKKR and German MAKEI 96/MHO 98. I think most of the patients in the two dataset should be Germany. There should not be such a big difference. Is there any bias during data collection?
2. Another concern is about the reference population. Many early onset cancers are related to genetic mutations. When selecting reference population, a population of similar genetic background as the study population should be used. "Orphanet was established in France by the INSERM (French National Institute for Health and Medical Research) in 1997. This initiative became a European endeavour from 2000, supported by grants from the European Commission: Orphanet has gradually grown to a Consortium of 40 countries, within Europe and across the globe." The reference population is not very similar to the study population, which may influence the estimation of OR in this paper. Similar issue for "Mainz" data.
3. Figure 1 and 2. What are the numbers in the figure. It should be explained either in figure legend or main text.
4. Table 1b, I am confused about malignant tumor column. "Severe CHD 1 f 12 years 2/10". Why is the malignant tumor column is 2/10 here.
Comments on the Quality of English Language
NA
Author Response
Reviewer 1
A major concern is about the study population, German MAKEI 96/MHO 98. "According to data from the DKKR (Deutsches Kinderkrebsregister) from 1987 to 2011, the incidence rate is 4.8 per million (children aged < 15 years). For girls, the incidence is 5.3, and for boys, 4.4 per million (2)". The difference between boys and girls are not very large. However, in the study population, "Of the 2,610 patients with an eGCT, 754 were male (28.9 %) and 1856 were female (71.1 %). " Almost 3 times difference between girls and boys. Why is there such a big difference between DKKR and German MAKEI 96/MHO 98. I think most of the patients in the two dataset should be Germany. There should not be such a big difference. Is there any bias during data collection?
Thank you for this important question. Concerning the different incidences between the data of the DKKR and MAKEI 96 one explanation for the difference in the data is that the DKKR included patients up to the age of 15 and the MAKEI study up to the age of 19. Ovarian tumors in particular occur in adolescence after the age of 15. Furthermore teratomas were not included in DKKR. Overall, in both registries the incidence of girls is higher then the incidence for boys.
Another concern is about the reference population. Many early onset cancers are related to genetic mutations. When selecting reference population, a population of similar genetic background as the study population should be used. "Orphanet was established in France by the INSERM (French National Institute for Health and Medical Research) in 1997. This initiative became a European endeavour from 2000, supported by grants from the European Commission: Orphanet has gradually grown to a Consortium of 40 countries, within Europe and across the globe." The reference population is not very similar to the study population, which may influence the estimation of OR in this paper. Similar issue for "Mainz" data.
To explain the choice of the reference population we want to remind that we perform a case-control study. The principle of the case-control study is to compare the prevalence of cases (MAKEI 96/MAHO 98) with or without genetic syndromes to controls from the general population. Thus the Orphanet data base describes the prevalence of genetic syndromes in the general population. Since both cases and controls arise from the general population a case-control approach is valid. In the text we chose the following amendment: “The principle of the case-control study is to compare the prevalence of cases (MAKEI 96/MAHO 98) with or without genetic syndromes to controls from the general population.” (https://www.orpha.net/) (lines 159-160)
The same applies to the comparison of the MAKEI 96/ MAHO 98 patients to cases registered in the Mainz registry, which is used as a control reflecting congenital anomalies within the general population. Here it might be argued that the Mainz population is not representative for Germany; although the site of Mainz was not selected at random, there is no reason to believe that the Mainz region is representative. But the undoubted strength of the Mainz registry is completeness and precision because of elaborate case definitions in respect to congenital anomalies. Therefore, we chose in the text the following wording: “In the Mainz registry cases are highly likely to reflect the general population.” (Queißer-Luft A, Spranger J. Fehlbildungen bei Neugeborenen. Dtsch Arztebl. 2006; 103(38): A-2464 / B-1174 / C-1125) (lines 165-166)
Figure 1 and 2. What are the numbers in the figure. It should be explained either in figure legend or main text.
The numbers in the figures are now explained in the figure legend (lines 232-233 and 265-266)
Table 1b, I am confused about malignant tumor column. "Severe CHD 1 f 12 years 2/10". Why is the malignant tumor column is 2/10 here.
The legend of table 1b was edited (lines 253-254):
Table 1b. Germ cell tumor divided by co-occurring congenital anomalies and tumor location (number of cases, sex, age at diagnosis and dignity (number of malignant tumors of all tumors belonging to a localization)).
Reviewer 2 Report
Comments and Suggestions for Authors
Childhood extracranial germ cell tumors are fairly rare tumors that form from germ cells (which are cells that can develop into sperm or eggs) in those parts of the body different from the brain. The causes are mostly unknown. They can be either benign or malignant, and are typically separated into gonadal (occurring in the testicles or ovaries) or extragonadal (occurring outside the gonads). This study searched for potential strong associations between syndromes or congenital anomalies and extracranial germ cell tumors. Such strong effects were observed for for Swyer and Currarino syndrome, but in the case of Klinefelter, Turner and Down syndrome the strength of risk appeared to be substantially low.
Overall, this is a good statistical survey of available patient data. I have a few minor observations only:
Line 73 - please provide more keywords than just the three you have listed
Line 69 - while in this case they may have all pertained to girls, this should not be overemphasized in the abstract or conclusions, since you did not perform any statistics separating the group of males from that of females or an initial principal component analysis that would identify patient sex as a principal component. You may want to perform these in order to strengthen your claim or instead refrain from such categorical wording.
Introduction - while there is data from literature about the incidence of such tumors, please add data from literature about the mortality also
Throughout the manuscript : square brackets, not round brackets, should be used for in-text citing of numbered references from the list at the end of the paper
Sentence on lines 331-332 is not clear. Please reword to clearly convey the message you want to convey.
Reference list must be revised: journal names must be abbreviated, volume number italicized, and please make sure you add page numbers or article number for ALL references listed.
Author Response
Reviewer 2
As additional keywords I chose: Developmental tumors, Swyer syndrome, Currarino syndrome, MAKEI 96/MAHO 98.
Line 69 - while in this case they may have all pertained to girls, this should not be overemphasized in the abstract or conclusions, since you did not perform any statistics separating the group of males from that of females or an initial principal component analysis that would identify patient sex as a principal component. You may want to perform these in order to strengthen your claim or instead refrain from such categorical wording.
Thank you for this suggestion. We added a calculation of the prevalence of female patients among the syndromes in comparison to the general population. 16 new syndromes were found in children with eGCTs and most of them were female (75%, n = 12). On average, the prevalence for girls in the general population was 42% for the syndromes mentioned (https://www.orpha.net/). (lines 234-244)
Introduction - while there is data from literature about the incidence of such tumors, please add data from literature about the mortality also
Data from literature about the mortality was included in the introduction.
In a study from 2023, the 5-year survival rate (EFS) and overall survival were 82.33% (95% CI, 77.32%, 86.62%) and 94.13% (95% CI, 90.02%, 96.69%) respectively (lines 83-84)
(Jiang S, Dong K, Li K, Liu J, Du X, Huang C et al. Extracranial Germ Cell Tumors in Children: Ten Years of Experience in Three Children's Medical Centers in Shanghai. Cancers (Basel) 2023 Nov 14;15(22):5412. doi: 10.3390/cancers15225412.)
Throughout the manuscript : square brackets, not round brackets, should be used for in-text citing of numbered references from the list at the end of the paper
Reference list must be revised: journal names must be abbreviated, volume number italicized, and please make sure you add page numbers or article number for ALL references listed.
Thanks for the advices, we edited the brackets with the references and the reference list.
Sentence on lines 331-332 is not clear. Please reword to clearly convey the message you want to convey.
The sentence was reworded: For other syndromes and non-syndromic defects this may be less evident, but cannot be excluded. (now lines 346-347).